# Genetic and Antiviral Potential Characterization of Four Insect-Specific Viruses Identified and Isolated from Mosquitoes in Yunnan Province

**DOI:** 10.3390/v17050596

**Published:** 2025-04-23

**Authors:** Qinxuan Miao, Lulu Deng, Xiang Le, Qian Li, Yuting Ning, Yimeng Duan, Qi Liu, Yinzhu Tao, Binghui Wang, Xueshan Xia

**Affiliations:** 1Faculty of Life Science and Technology, Kunming University of Science and Technology, Kunming 650500, China; qinxuanmiao@aliyun.com (Q.M.); 3055779157@aliyun.com (L.D.); lexiang1936@163.com (X.L.); contrinalq@126.com (Q.L.); nyt31894601@126.com (Y.N.); dym17864170070@126.com (Y.D.); liuqi2401@163.com (Q.L.); 2Yunnan Province Key Laboratory of Public Health and Biosafety, School of Public Health, Kunming Medical University, Kunming 650500, China; yinz_14@163.com

**Keywords:** genetic, pathogenic, insect-specific viruses, mosquito-borne viruses

## Abstract

Mosquitoes, comprising over 300 species, are pivotal vectors for transmitting arthropod-borne viruses (arboviruses) to vertebrates via bites, posing a significant public health threat with approximately 700,000 annual deaths. In contrast, insect-specific viruses (ISVs) exclusively infect insects and have no direct impact on human health. Yunnan Province in China, located in tropical and subtropical regions, provides an ideal environment for mosquito habitation and has the highest diversity of known mosquito-borne viruses. In this study, mosquito samples were collected from eight cities and states in Yunnan Province, totaling 15,099 specimens. Based on the collection sites and mosquito species, the samples were divided into 110 groups for virus isolation. Four insect-specific viruses (Tanay virus [TANV], Culex orthoflavivirus [CxFV], Aedes orthoflavivirus [AeFV], La Tina virus [LTNV]) were successfully isolated, and co-infection studies with dengue virus (DENV-2) were conducted in C6/36 cells. Preliminary results suggested that these four insect-specific viruses may reduce the viral titer of DENV-2 in C6/36 cells. Understanding the intricate interactions between insect-specific viruses and mosquito-borne viruses is crucial for elucidating the multifaceted role of mosquitoes in arboviral transmission dynamics. Insect-specific viruses exhibit considerable potential as innovative biocontrol agents, with promising capacity to attenuate mosquito-borne viral transmission through the targeted modulation of mosquito innate immunity and physiological adaptations.

## 1. Introduction

Mosquitoes serve as vectors for a wide array of viruses, and mosquito-borne viruses are a major contributor to the global emergence and re-emergence of infectious diseases, thereby posing a substantial public health risk worldwide. Mosquito-borne viruses can be broadly classified into two categories: arthropod-borne and insect-specific viruses (ISVs). Arthropod-borne viruses are transmitted to humans and animals through mosquitoes and other arthropods. Common examples of arthropod-borne viruses include dengue virus, Zika virus, and Japanese encephalitis virus. ISVs can replicate only in insect hosts and generally pose no threat to mammals or humans [1,2]. ISVs can co-infect the same host with other pathogens, such as bacteria, fungi, or other viruses, potentially influencing the pathogenicity, transmission efficiency, and host’s immune response [3]. Insect-specific viruses (ISVs), such as Culex orthoflavivirus (CxFV), can upregulate host RNA interference (RNAi) components (e.g., Dicer-2 and Argonaute-2), thereby enhancing the degradation efficiency of co-infected arboviruses like dengue virus (DENV) through the RNAi pathway. Additionally, certain ISVs encode viral miRNAs that specifically target and silence host cuticle protein genes, consequently inhibiting arbovirus budding [4]. The discovery of these novel viruses has sparked interest in their potential applications as biological control agents and platforms for novel vaccine development [1].

Insect-specific orthoflaviviruses (ISFs) differ significantly from medically important flaviviruses, such as the dengue virus and West Nile virus (WNV), in terms of their virology, transmission mechanisms, and host interactions. Although ISFs do not infect mammals or humans, they play a complex role in the ecology of mosquito hosts and viral transmission, potentially influencing the transmission dynamics of medically important flaviviruses. ISFs may inhibit the replication of medically significant flaviviruses by sharing the same host cells or utilizing similar biological pathways [5,6,7,8]. For example, ISFs may reduce the replication capacity of medically significant flaviviruses by competing for host cell resources such as enzymes and transcription factors required for viral RNA synthesis [5,9,10,11]. Some ISFs may indirectly promote the replication of medically significant flaviviruses by modulating the host immune system or intracellular environment [11,12,13,14].

## 2. Materials and Methods

This study was conducted from July 2021 to July 2024 in major mosquito distribution areas of Yunnan Province, China. Mosquito sampling was performed in high-density habitats, particularly near rubber plantations and bamboo forests, where mosquito samples were collected using human-baited and trap methods. Mosquito collections were conducted monthly during peak activity seasons (July–October) from 2022 to 2024, with sampling focused on crepuscular periods (06:00–09:00 and 18:00–21:00 local time) to maximize the capture efficiency of host-seeking females. Captured mosquitoes were euthanized by rapid freezing in a dry ice chamber, followed by morphological identification under a stereomicroscope. Species identification was confirmed by DNA barcoding of the mitochondrial cytochrome c oxidase subunit I (COI) gene following RNA extraction and reverse transcription. All samples were stored at −80 °C. Mosquito specimens (*n* = 15,099) were stratified by geographic origin (8 collection sites), sampling timepoint (July 2021 to July 2024), and species identification (morphological and COI barcoding confirmation). From each stratum, 40–45 individuals were randomly selected to form 110 pooled samples, ensuring representative coverage of spatiotemporal and entomological variables. Whole mosquito bodies (including head, thorax, abdomen, and appendages) were homogenized in viral transport medium using a Precellys 24 homogenizer; viral RNA was extracted using the TRIzol reagent. Viral RNA screening was performed by RT-PCR using sequence-specific primers targeting conserved viral genomic regions (see Appendix A for primer details). The RT-PCR-positive samples were inoculated into C6/36 cells, and cytopathic effects (CPEs) were observed (the GenBank accession numbers for all sequenced strains and corresponding sample metadata are provided in Appendix A). Viral quantification was performed using RT-qPCR to measure the RNA copy numbers of DENV-2 and other viruses (TANAV, LTNV, CxFV, and AeFV). TANAV and LTNV were selected at random for the experiment (see Appendix A for fluorescent quantitative primer probe information). DENV-2 was titrated in BHK-21, and TANAV, LTNV, CxFV, and AeFV were titrated in C6/36 cells using the plaque assay.

TANV primers were designed to amplify the full-length genome of the successfully isolated viruses, and a phylogenetic tree based on the full-length sequences was constructed (detailed primer information is provided in Appendix A). The positive samples of AeFV, CxFV, and LTNV were pooled for next-generation sequencing (NGS), which was performed at Magigene, and the sequencing data obtained were assembled from scratch using an automated pipeline. To validate the sequences obtained and to obtain the complete genome, primers were designed using Oligo 7. These primers were used to validate and amplify the viral genome. Phylogenetic analysis was performed using MEGA-X_V10.2.6 software. Full-length RNA sequences were submitted to the NCBI for collation in the Biotechnology Information GenBank database. TANAV (accession number: PV165524), LTNV (accession number: PV165523), CxFV (accession number: PV165384), and AeFV (accession number: PV165522). Sequences were aligned with MAFFT. Maximum likelihood analysis was conducted using MEGA-X, with a general time-reversible model (GTR + G + I) and 1000 bootstraps.

## 3. Results and Discussion

Yunnan Province, located in southwestern China, represents a critical region for arbovirus research due to its unique geographical position and ecological diversity. Characterized by tropical and subtropical climates, the warm and humid environment provides ideal conditions for mosquito proliferation and pathogen transmission. This ecological setting not only facilitates the maintenance and spread of existing mosquito-borne viruses but may also contribute to the emergence and evolution of novel arboviruses. In this study, 15,099 mosquito samples were collected from five species across eight regions of Yunnan Province (Xishuangbanna, Dehong, Honghe, Wenshan, Pu’er, Zhaotong, Chuxiong, and Lincang). The mosquito species included *Anopheles sinensis Wiedemann*, *Armigeres subalbatus*, *Aedes albopictus*, *Aedes aegypti*, and *Culex tritaeniorhynchus Giles* (Figure 1). Based on the collection sites and mosquito species, the samples were divided into 110 groups for virus isolation. Fifteen groups of samples exhibited CPE after virus isolation from C6/36 cells, and CPE was consistently observed in each subsequent passage. Using specific universal primers to screen mosquito-derived isolates for CPE, we successfully isolated and identified seven strains of TANAV with a CPE positivity rate of 70% (7/10). Additionally, one strain of CxFV and one strain of AeFV were isolated, both demonstrating a CPE positivity rate of 100% (1/1). Six strains of LTNV were obtained, with a CPE positivity rate of 100% (6/6). These findings indicate that TANAV, CxFV, AeFV, and LTNV exhibit high isolation rates in mosquito samples and induce significant CPE in the collected samples. In this study, TANAV was successfully isolated from multiple mosquito species, including *Aedes albopictus*, *Aedes aegypti*, *Culex tritaeniorhynchus*, *Armigeres subalbatus*, and *Anopheles sinensis*. In contrast, LTNV was isolated exclusively from *Aedes albopictus*. These findings suggest that TANAV exhibits a broad host range, whereas LTNV may demonstrate higher host specificity.

Phylogenetic analysis based on full-length genome sequences revealed that the Tanay virus strain isolated in this study exhibited the closest genetic relationship with the YN15-103-01 strain, which was isolated from *Anopheles sinensis* in Yunnan Province in 2015, with nucleotide sequence similarities ranging from 98.68% to 99.23%. These findings support the genetic diversity of TANAV in the Yunnan region and highlight its high homology with historical isolates (Figure 2a). Phylogenetic analysis based on full-length genome sequences revealed that the CxFV isolated from *Culex tritaeniorhynchus* in Pu’er City, Yunnan Province, formed a distinct branch in the phylogenetic tree and showed a close genetic relationship with the H0901 strain isolated from Liaoning Province in 2010, with a sequence similarity of 92.11% in the envelope full-length genome sequences. These results suggest that CxFV strains from Pu’er may exhibit unique evolutionary characteristics while maintaining certain genetic connections with strains from northeastern China (Figure 2c). Phylogenetic and antigenic analyses revealed that the La Tina virus exhibits minimal divergence from classical ISFs, such as cell-fusing agent virus and Culex orthoflavivirus. Further genomic characterization demonstrated that our isolated strain shares 99% nucleotide identity in the NS5 gene with the reference strain 49-LT96 (GenBank accession: KY320649), indicating close evolutionary relatedness. (Figure 2b). Phylogenetic analysis revealed that Aedes orthoflavivirus (AEFV) forms a distinct clade with Culex orthoflavivirus (CxFV) and Quang Binh virus (QBV), which is evolutionarily separate from medically important Aedes-associated orthoflaviviruses such as dengue virus (DENV) and West Nile virus (WNV). Further genomic characterization demonstrated that our isolate shares 99.12% nucleotide identity in the NS5 gene with the reference strain TC4A8_18-9L-Y-T-Aea-B-1-1 (GenBank accession: MT254427.1) from Tengchong, Yunnan Province. (Figure 2d). From a comprehensive phylogenetic perspective, LTNV, CxFV, and AeFV all belong to the Flaviviridae family and cluster within the same major evolutionary branch as the dengue virus. In contrast, TANAV is taxonomically unclassified and phylogenetically grouped within the *Negevirus* cluster, a recently proposed taxon comprising viruses isolated from hematophagous insects. Notably, all these viruses are classified as ISVs, highlighting their unique evolutionary adaptations to host restriction.

CxFV genotypes were classified as GI and GII [15]. The CxFV strain detected in *Culex tritaeniorhynchus* from Pu’er, Yunnan, was phylogenetically positioned between GI and GII based on E gene analysis. GI comprises the China, Japan, and USA groups, with approximately 5% nucleotide divergence between the groups. The E gene of the detected CxFV showed 92.38% similarity to the DG104 strain from *Culex pipiens pallens* in Liaoning, with >6% nucleotide divergence, suggesting the potential emergence of a new genotype within the Chinese population [16].

In this study, LTNV and AeFV exhibited 98% nucleotide similarity; both are dual-host-associated dISFs that are phylogenetically closer to vertebrate-pathogenic *Flaviviridae* [16]. Additionally, dISFs are closely related to flavivirus pathogens, such as WNV, Zika virus, and dengue virus. These similarities suggest that dISFs may modulate arbovirus infection and transmission in dually infected mosquito hosts or can be used to develop flavivirus vaccines and as diagnostic tools [17].

The nucleotide similarity between TANAV isolated in this study and the 2015 Yunnan Anopheles isolate (YN15-103-01) [14] was 98.68–99.23%; similarities to TANAV from the Philippines [10], Guangxi, and China were 79% and 85%, respectively. YN15-103-01 and TANAV isolated in this study belong to *Negevirus*, a new taxon within the TANAV group, including ISVs, such as Negev virus, Goutanap virus, Bustos virus, and Santana virus. Previous studies [14] have indicated that *Negeviruses* are predominantly distributed between 42° N and 42° S in tropical and subtropical regions, suggesting that their distribution is influenced by environmental and climatic factors. Further insight into TANAV and related ISVs is expected to provide valuable insights into their biological characteristics and ecological distributions.

Consistent with our survey data, Yunnan’s diverse climatic zones exhibit distinct mosquito species assemblages: *Aedes* spp. and *Armigeres subalbatus* dominate tropical ecotones (e.g., rubber plantations and bamboo forests), while *Culex tritaeniorhynchus* and Anopheles sinensis prevail in peri-domestic livestock enclosures. Although the dengue virus has not established stable local transmission in China, the widespread distribution of its vector, *Aedes* mosquitoes, has led to annual local outbreaks, with epidemic scales showing a cyclical upward trend influenced by global dengue dynamics. Yunnan Province, located in southwestern China and bordering Myanmar, Laos, and Vietnam, reported imported dengue cases annually from 2013 to 2024, alongside multiple local outbreaks. In 2023, Yunnan reported 13,482 cases of dengue, accounting for 68.99% of the national total. To investigate the impact of ISVs on dengue virus carriage in mosquitoes, we co-infected C6/36 cells with four ISVs and DENV-2 (Figure 3). Co-infection with ISVs significantly reduced DENV-2 titers, albeit to varying degrees. TANAV exhibited the most pronounced suppression of DENV-2 titers, whereas LTNV and AeFV showed milder and comparable effects. CxFV demonstrated an intermediate suppression between TANAV and LTNV/AeFV. In C6/36 cells co-infected with TANAV/DENV, DENV-2 titers were significantly reduced, with a mean peak titer of 2.08 log10 PFU/mL. Mean peak titers in LTNV/DENV and AeFV/DENV co-infected cells were 3.65 log10 PFU/mL and 3.85 log10 PFU/mL, respectively. CxFV/DENV co-infection resulted in a mean peak titer of 3.19 log10 PFU/mL. In contrast, DENV-2 mono-infected controls exhibited a significantly higher mean peak titer of 5.55 log10 PFU/mL.

TANAV, LTNV, CxFV, and AeFV reduced DENV-2 titers in C6/36 cells. LTNV, AeFV, and CxFV, which are typical ISFs, exhibited suppressive effects, consistent with previous reports. Notably, TANAV, belonging to the *Negevirus* group (a recently proposed taxon of insect-specific viruses), significantly suppressed DENV-2 titers, with cell kinetic studies indicating a more pronounced inhibitory effect. While prior studies have only reported DENV-2 suppression by ISFs within the Flaviviridae family, our findings revealed, for the first time, that non-flavivirus ISVs (e.g., TANAV) may also possess such capabilities.

Our preliminary study revealed that TANAV, LTNV, CxFV, and AeFV variably reduced DENV-2 titers in C6/36 cells. Further research is required to explore the potential mechanisms underlying the interactions between these ISVs and the dengue virus. Additionally, limitations, such as the possibility that these viruses and DENV may not infect the same mosquito species, merit further consideration. While our study examined simultaneous co-infection, prior work suggests that sequential infection timing critically impacts interference outcomes—ISV pre-infection typically suppresses arboviruses, whereas established arbovirus infections may exclude secondary ISVs. This temporal regulation warrants dedicated investigation.

## Figures and Tables

**Figure 1 viruses-17-00596-f001:**
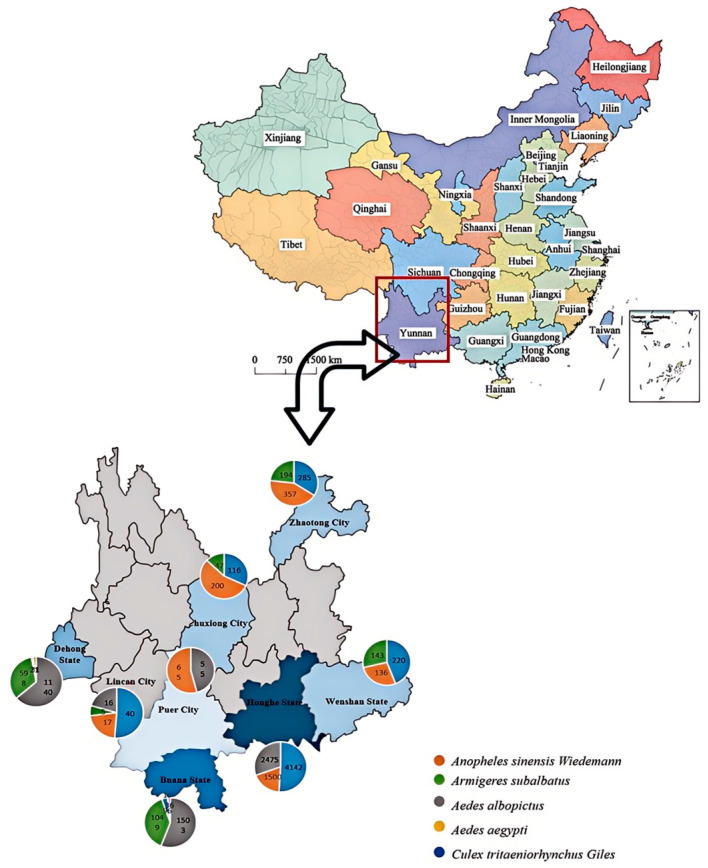
Geographic distribution of mosquito sampling sites in Yunnan Province, China.

**Figure 2 viruses-17-00596-f002:**
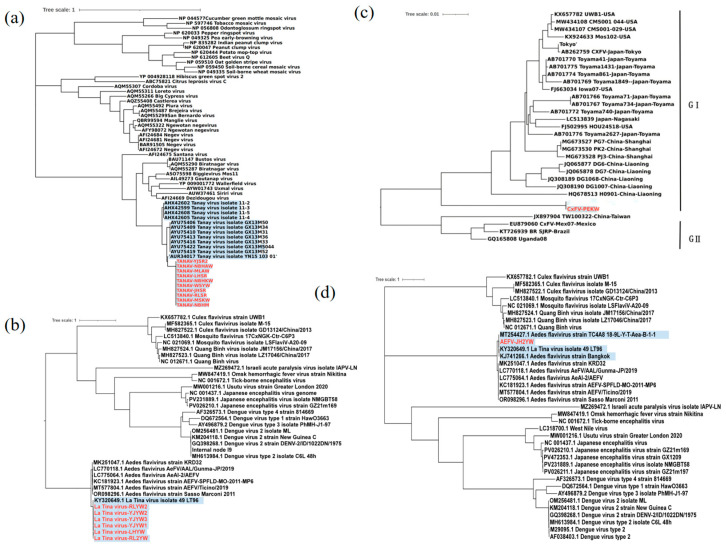
(**a**–**d**) Phylogenetic analysis based on full-length genomes of TANAV, LTNV, CxFV, and AeFV. Blue color indicates strains with high similarity to strains found in this study and clustered together in the evolutionary tree, red color represents strains obtained in this study.

**Figure 3 viruses-17-00596-f003:**
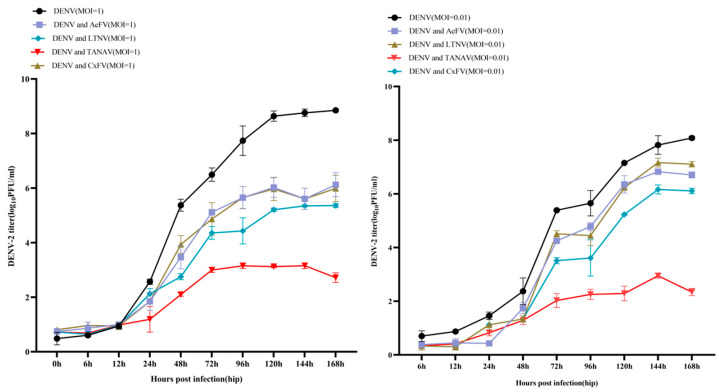
TANAV, LTNV, CxFV, and AeFV reduce viral titers of DENV-2 in C6/36 cells (MOI = 1/MOI = 0.01).

## Data Availability

All the relevant data are included in the manuscript in an aggregated form.

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
