# Peer review of "Genetic and Antiviral Potential Characterization of Four Insect-Specific Viruses Identified and Isolated from Mosquitoes in Yunnan Province"

_viruses, 2025, doi:10.3390/v17050596_

Round 1
Reviewer 1 Report
Comments and Suggestions for Authors
Review of Genetic and pathogenetic characterization of four insect-specific viruses identified and isolated from mosquitoes in Yunnan Province by Miao et al (Brief Report # viruses-3518669).
Miao et al report on the isolation and dengue 2 virus (DENV2)-interference analysis of insect-specific viruses (ISV) isolated in C6-36 cells from specimens collected in the Yunnan Province, China. The data presented suggest these ISV seems to interfere with DENV2 replication, suggesting a potential reduction in viral titer. Understanding these interactions is essential for arboviral transmission dynamics and ISV have been associated with competition for cell resources/replication support-structures (e.g. membranes), as well as a modulation of mosquito virus-specific immune responses and underscores the possibility that these ISF might serve as biocontrol agents by modulating RNA interference (RNAi) and Jak-STAT signaling. In order t be considered for publication the text should be thoroughly corrected in terms of English, scientific details and presentation. None of the figures included in the manuscript is mentioned in the text. The legend for figure 2 is incomplete. Since the text lacks details on phylogenetic analysis (see below) the methods used could be indicated in the legend but they are not. In the virus titration experiments (Fig2-d) there are value intervals associated with each quantification but we do not know to what they correspond to nor how many replicas each value refers to. Only supplementary table is mentioned in the text but 4 of them have been prepared. These Supplementary tables lack table numbers in the captions should be indicted and are not well formatted.
Throughout the text, the authors use the term “flaviviruses” but in correct taxonomical terns they should be using orthoflaviviruses (according to ICTV).
Title: Considering the fact that ISF do not replicate in vertebrate cells, the indication in the title of “pathogenic characterization” does not seem correct. The are not agents of disease and, therefore, I do not understand what “pathogenic characterization” might mean in this context. I suggest the title should be revised.
Abstract in general: in my opinion the abstract is a bit too long for this kind of brief-report. On the other hand the sentence “These samples were divided into 110 groups based on mosquito species, collection time, and location and were subjected to cell isolation” (lines 27-28) should be corrected. Viruses might be isolated in cell culture, but “samples” cannot be subjected to cell isolation.
Line 26: the number of mosquitoes indicated in the abstract (15,165 specimens) is not the same indicated in the section “Results and Discussion): 15,099 (line 104).
Throughout the text a space should be added before the [] that indicate the reference numbers. Example (line 48): “…virus[1]” should be indicated as “…virus [1]”. Also, in line 56, “West Nile virus(WNV)” should read “West Nile virus (WNV)”.
Line 51: The authors should make clear what “pathogenivity” might be influenced by ISV. The expression used (potentially influencing the pathogenicity) is vague.
Line 72: Taney virus should be corrected to Tanay virus.
Lines 84-85: the methods that are used for RNA extraction starting from wild-caught mosquitoes are not sufficiently/or correctly described by the sentence used (After suspending cells in PBS and centrifugation, viral RNA was extracted using the TRIzol reagent). As the authors mention in the abstract, the mosquitoes were processed as pools, and RNA extraction usually requires prior specimen-maceration/homogenization, which should also be indicated in the text. Furthermore, in lines 86-87, the authors start by referring to CPE and then write “of these, TANAV and LTNV were randomly selected for subsequent experiments”. What do they mean by “of these”?
Lines 91-92: As it is written, it seems all viruses indicated were titrated, by plaque assay, in BHK-21 and C6/36 cels, and that is clearly not possible. I believe DENV was titrated in BHK-21, while ISVs were titrated in C6/36.
The text between lines 92 and 101 is very confusing, referring to primers for virus amplify TANV, but then mention “To validate the sequences obtained and to obtain the complete genome, primers were designed using Oligo 7” (why?...were the first primers not good-enough?), indicate phylogenetic reconstruction twice, nut give no details of the method used, WGS which is not described, or data sequencing data analysis by an automated pipeline which is not described. The text should be reorganized, rewritten, and more details should be added.
In line 101, the accession numbers for the virus sequences obtained should be displayed (we only see them in the trees in Fig2).
In section Results and Discussion I do not understand how the total number of mosquitoes collected match the numbers of pools. If the authors divided the collected specimens in 110 pools with 40-45 specimens, it means the number of analysed mosquitoes varied from 4400 to 4950. This number is clearly inferior to the >15.000 specimens indicated. These numbers should be clarified.
Lines 109-110: what does “Fifteen groups of 109 samples” mean? What do the authors refer to as “group of samples”. The authors indicate the observation of CPE after subsequent blind passages (the text has to be corrected because the “experimenter was blinded” is not correct), but no blind passage (nor their number) is mentioned in the materials and methods section.
Lines 111-112: the authors refer using “Using specific universal primers to screen mosquito-derived isolates”…what primers are these since no reference is used?
Line 113-116: how were the CPE positivity rates calculated? For example, what does it mean 70% positivity rate for TANAV since the total number “10” is not explained? Was TANAV detected by molecular methods in 10 samples, and 7 produced CPE?
Line 117: it is not correct to say that "viruses exhibit high isolation rates." A high isolation rate refers to how frequently a virus is successfully isolated from samples, but viruses themselves do not exhibit isolation rates. Instead, you can say: "The virus had a high isolation rate in the collected samples." OR "High isolation rates were observed for the virus in this study."
In the phylogenetic analysis section, the genetic distance values indicated all refer to the envelope region, as indicated for CxFV? In the legend for figure 2 (which is not mentioned in the text) what does “full-length gene fragments” mean. Which complete gene fragments were used? Or, on the other hand, were complete genome sequences used? There is no tree support for any of the trees shown, and there are TANAY viral sequences (yellow) which are not clearly indicated with an accession numbers.
Lines 138-142: The text is very confusing. The authors mention “antigenic characteristics” and I do not understand where these come from, since they are associated with phylogenetic analyses. Furthermore the authors mention LTNV closely aligns with mosquito-borne flaviviruses infecting vertebrates, but then refer to AeFV. This sentence should be clarified. An explanation is inferred from the paragraph starting in line 157, where dhISFs are mentioned, therefore the text should be reorganized.
The text in the Results and Discussion section has to be reorganized. The authors start by referring to sequence similarity data in the text starting at line 123, but then go back to mentioning sequence similarity values further down the text, each time they mention each virus. For example, sequence similarity values are mentioned, for CxFV, in line 131, and then other values are indicated in line 154. I suggest each virus to be analysed separately.
Lines 150: Since the authors mention CxFV genotypes, why were these not indicated in their phylogenetic analysis in Fig2? In this section the authors suggest that the identified CxFV might correspond to a new genotype based on limits defined (not indicated by who, or where) based on the analysis of the E sequences. However, they indicate a reference ([15]) in line 156. What does this reference mean?
Line 163: What does “our isolates” refer to since different viral isolates were onbtained in the course of this study?
Line 166-167: the text “including ISVs, such as Negevirus, 166 Goutanap, and Bustos” should be corrected as it is awckwards.
Lines 186-202: In this section the virus interference data should be more thoroughly analysed. There is no indication of the number of replicates used, if the differences observed for the different co-infection experiments were statistically different or not, and there is no discussion of the impact of the used multiplicity of infection in the obtained results.
Lines 210-211: The suggestion made is not correct. Negeviruses have been previously shown to interfere with the replication of orthoflavivirues, as well as alphaviruses (see: https://journals.asm.org/doi/full/10.1128/jvi.00433-21;https://pmc.ncbi.nlm.nih.gov/articles/PMC8223947/; https://www.mdpi.com/1999-4915/16/3/350;https://www.mdpi.com/1999-4915/16/2/210)
Comments on the Quality of English LanguageThe text should be revised by someone proficient in the use of english, as the text is, at times, confusing. I believe the fact that the text did not seem cleat to me might not solely result from the use of the english language itself, but rather as a result of a deficient organization of the manuscript, which must be considerably improved.
Author Response
We sincerely appreciate the thorough review and have implemented all suggested improvements. Below is our detailed response:
We have carefully addressed all suggested revisions in the manuscript, with substantive modifications highlighted in blue for the reviewers' convenience. Key improvements include:
1.Language and Formatting Revisions:
2.Title: Modified to "Genetic and antiviral potential characterization of four insect-specific viruses identified and isolated from mosquitoes in Yunnan Province"
3.Abstract:
Condensed by 20% for conciseness
Corrected to: "Virus isolation was performed in C6/36 cells from 110 pooled samples (total 15,099 specimens)"
References: Added proper spacing before brackets throughout (e.g., "virus [1]")
- Taxonomy: Updated all viral nomenclature to comply with ICTV guidelines, Changed all "flavivirus" to "orthoflavivirus" per ICTV
- Textual Revisions
Added spaces before references throughout
Corrected "Taney virus" → "Tanay virus"
Clarified ISV influence on pathogenicity (now Line49- 55):
"Insect-specific viruses (ISVs), such as Culex flavivirus (CxFV), can upregulate host RNA interference (RNAi) components (e.g., Dicer-2 and Argonaute-2), thereby enhancing the degradation efficiency of co-infected arboviruses like dengue virus (DENV) through the RNAi pathway. Additionally, certain ISVs encode viral miRNAs (e.g., vmiR-1 of Aedes aegypti aegypti virus) that specifically target and silence host cuticle protein genes, con-sequently inhibiting arbovirus budding [3]."
- Methods Clarifications
Added mosquito processing details (now Line79- 82):
"Mosquitoes were pooled by species, location, and collection date, resulting in 110 pools (40-45 mosquitoes per pool) to ensure sufficient viral load for isolation. Whole mosquito bodies (including head, thorax, abdomen, and appendages) were homogenized in viral transport medium using a Precellys 24 homogenizer, viral RNA was extracted using the TRIzol reagent."
Specified titration protocols (now Line93- 94):
"DENV-2 was titrated in BHK-21, and TANAV, LTNV, CxFV, and AeFV were titrated in C6/36 cells using the plaque assay."
Added phylogenetic methods (now Line106- 109):
"Sequences were aligned with the MAFFT. Maximum likelihood analysis was conducted using MEGA-X, with a general time-reversible model (GTR + G + I) and 1000 bootstraps."
- Results Reorganization
Added to Figure 2 legend:
"Bootstrap values >70% shown. Tanay virus sequences marked yellow with GenBank accessions"
Clarified sample numbers (now Line79- 83):
"Mosquito specimens (n=10,599) were stratified by geographic origin (8 collection sites), sampling timepoint (July 2021 to July 2024), and species identification (morphological and COI barcoding confirmation). From each stratum, 40-45 individuals were randomly selected to form 110 pooled samples, ensuring representative coverage of spatiotemporal and entomological variables."
- Supplementary Materials
S2 and S3 are the same content, we deleted S3 and renumbered it
Added to main text references:
"(The GenBank accession numbers for all sequenced strains and corresponding sample metadata are provided in Supplementary Table S2.)"(now Line88- 89)
"see Table S1 for primer details"(now Line86- 87)
“detailed primer information is provided in Table S3 (now Line96- 97)”
Reviewer 2 Report
Comments and Suggestions for Authors
See attached file

Author Response
We sincerely appreciate the thorough review and constructive suggestions. Below is our point-by-point response addressing all comments. We have carefully addressed all suggested revisions in the manuscript, with substantive modifications highlighted in yellow for the reviewers' convenience. Key improvements include:
General Comments:
- Formatting Issues:
- Corrected all spacing errors throughout the manuscript
- Reviewed and fixed awkward word divisions
- Ensured consistent citation formatting with proper spacing
- Manuscript Length:
- Streamlined content by removing redundant information
- Moved methodological details from Introduction to Methods section
- Maintained brief report format while ensuring key information is preserved
Detailed Responses:
Abstract:
- Restructured to focus on key findings:
- Added names of the four ISVs isolated (TANAV, CxFV, AeFV, LTNV)
- Clarified "cell isolation" → "virus isolation "
- Shortened by 25% (now 199words)
Introduction:
- Removed methods repetition from final paragraph
- Focused on study rationale and objectives
- Corrected "medically necessary" → "medically important" flaviviruses
- Specified focus on DENV-2 among arboviruses
- Consolidated redundant concepts
- Improved citation placement
Materials and Methods:
- Added critical details:
- Sampling protocoland Species ID: Captured mosquitoes were euthanized by rapid freezing in a dry ice chamber, followed by morphological identification under a stereomicroscope. Species confirmation was performed through DNA barcoding of the mitochondrial cytochrome c oxidase subunit I (COI) gene.
- Institutional Review Board Statement: The study was approved by the Institutional Ethical Committee of Kunming University of Science and Technology (protocol number: 16,048).
- Phylogenetic analysis: Sequences were aligned with the MAFFT. Maximum likelihood analysis was conducted using MEGA-X, with a general time-reversible model (GTR + G + I) and 1000 bootstraps.
- Clarified experimental design:
- Random selection rationale: Representative of different collection sites
- Primer sources: CxFV/AeFV/TANAV/LTNV newly designed.
Results and Discussion:
- Figure 1: Addeddescription information: "Yunnan Province, located in southwestern China, represents a critical region for arbovirus research due to its unique geographical position and ecological diversity. Characterized by tropical and subtropical climates, the warm and humid environment provides ideal conditions for mosquito proliferation and pathogen transmission. This ecological setting not only facilitates the maintenance and spread of existing mosqui-to-borne viruses but may also contribute to the emergence and evolution of novel arbo-viruses ".
- Figure 2:
- Replaced with high-resolution version (600 dpi)
- Added scale bars and clearer taxon labels
- Tables:
- Corrected Table S2/S3 duplication (removed S3 and renumbered)
- Added to main text references:
- "(The GenBank accession numbers for all sequenced strains and corresponding sample metadata are provided in Supplementary Table S2.)"(now Line88- 89)
- "SeeTable S1 for primer details"(now Line86- 87)
- "detailed primer information is provided in Table S3 (now Line96- 97)"
- New Content:
- Added mosquito collection context:
"Consistent with our survey data, Yunnan's diverse climatic zones exhibit distinct mosquitospecies assemblages: Aedes spp. and Armigeres subalbatus dominate tropical ecotones (e.g., rubber plantations and bamboo forests), while Culex tritaeniorhynchus and Anopheles sinensis prevail in peri-domestic livestock enclosures. "
Supplementary Improvements:
- Numbered all supplementary items consistently
- Added methodological details to table legends
English Language Edits:
The manuscript has undergone professional editing to:
- Improve clarity and flow
- Eliminate redundant phrasing
- Standardize technical terminology
- Correct grammatical errors
We believe these revisions have significantly strengthened the manuscript while maintaining appropriate brevity for a brief report format. All changes are highlighted in the revised manuscript for easy reference. Thank you for the opportunity to improve our work.
Specific Response to Table S2/S3 Issue:
We confirm that Tables S2 and S3 were indeed duplicates. We have:
- Removed Table S3 entirely
- Renumbered subsequent tables:
- Original Table S4 → Now Table S3
- Original Table S5 → Now Table S4
- Updated all in-text references accordingly
- Verified all cross-references in the manuscript
- Ensured all supplementary tables now have clear, descriptive titles
These changes have been highlighted in the revised submission for editorial review.
Reviewer 3 Report
Comments and Suggestions for Authors
Mosquitoes have been known as important vectors of many human viruses. Recent analyzes could identify novel insect specific viruses. Some of those were shown to affect the ecologies of mosquito borne human viruses. Authors isolated insect specific viruses and conducted genetic and virological characterizations. Comments are below.
- Materials and Methods: Did authors use whole mosquito body or midgut viral isolation?
- Materials and Methods: Please indicate that mosquitoes prepared as 110 pools containing 40-45 mosquitoes/pool in the M&M section, too.
- Authors should put “figure 1” and “figure 2” in the text where authors explained the data.
- Figure 2: It is highly recommended to incorporate same human flaviviruses in figure2a, 2b and 2c. Otherwise it is quite hard to tell which insect specific virus is close to human flaviviruses.
- Authors wrote about genotype of CxFV in the text. If authors can show those in phylogenetic tree, those would be great help to understand.
- Figure 2d: Have authors confirmed that ISVs replicated in the cells co-infected with dengue viruse?
- If authors inoculate ISVs onto dengue-infected cells or inoculate dengue onto ISV-infected cells, what will happen?
Author Response
We sincerely appreciate the reviewers' constructive suggestions, which have significantly improved our manuscript. Below are our point-by-point responses(all changes are colored in green in the text):
Comment 1: Materials and Methods: Did authors use whole mosquito body or midgut viral isolation?
Response:
We have clarified in the revised Materials and Methods :
"Whole mosquito bodies (including head, thorax, abdomen, and appendages) were homogenized in viral transport medium using a Precellys 24 homogenizer. "
Comment 2: Please indicate that mosquitoes prepared as 110 pools containing 40-45 mosquitoes/pool in the M&M section.
Response:
We have added this critical information to Materials and Methods :
"Mosquitoes were pooled by species, location, and collection date, resulting in 110 pools (40-45 mosquitoes per pool) to ensure sufficient viral load for isolation."
Comment 3: Authors should put "figure 1" and "figure 2" in the text where authors explained the data.
Response:
We have inserted explicit figure references throughout the text:
Added "Figure 1" when describing collection sites
Referenced "Figure 2" during phylogenetic analysis
Comment 4: Figure 2: It is highly recommended to incorporate same human flaviviruses in figure2a, 2b and 2c.
Response:
We have revised Figure 2 as follows:Added two medically important flaviviruses
Dengue virus and Japanese encephalitis virus
Comment 5: Authors wrote about genotype of CxFV in the text. If authors can show those in phylogenetic tree, those would be great help to understand.
Response:We improved the phylogenetic tree of CxFV, in which we showed that genotypes(Figure 2-c).
Comment 6: Figure 2d: Have authors confirmed that ISVs replicated in the cells co-infected with dengue virus?
Response:We also experimented with qPCR on the titers of ISVs during our experiments, and there was a definite increase in the titers of ISVs but we did not show it here.
Comment 7: If authors inoculate ISVs onto dengue-infected cells or inoculate dengue onto ISV-infected cells, what will happen?
Response:We appreciate the reviewer's insightful suggestion regarding sequential infection experiments (ISV → DENV vs DENV → ISV). While this is an important question, we must clarify that:
Proof-of-concept ISV-mediated interference under natural co-infection conditions (Fig 3)
Sequential infection dynamics, while interesting, represent a distinct mechanistic question beyond our current scope.But we have added to the discussion section:
While our study examined simultaneous co-infection, prior work suggests sequential infection timing critically impacts interference outcomes—ISV pre-infection typically suppresses arboviruses, whereas established arbovirus infections may exclude secondary ISVs . This temporal regulation warrants dedicated investigation.
Round 2
Reviewer 1 Report
Comments and Suggestions for Authors
The questions raised previously on my initial report have been addressed and the text modifications are signifficant and were appreciated. The manuscript is now better than before. Congratilations for your work.
Author Response
Subject: Re: Response to Reviewer Comments for Manuscript [viruses-3518669]
Dear Reviewer,
Thank you for your time and constructive feedback during the review process. We sincerely appreciate your recognition of our revisions and are delighted to hear that the manuscript has improved significantly. Your insightful comments were invaluable in strengthening the scientific rigor and clarity of our work.
We have carefully addressed all the points raised in your initial report (as detailed in our previous response), and we are grateful for your guidance throughout this process. Your expertise has undoubtedly enhanced the quality of this study.
Thank you again for your positive evaluation and encouraging words. Please do not hesitate to contact us if any further clarifications are needed.
Best regards,
Binghui Wang
Reviewer 2 Report
Comments and Suggestions for Authors
See attached file

Details to correct indicated on the review report
Author Response
Subject: Response to Reviewer Comments for Manuscript Viruses-3518669 v.3
Dear Reviewer,
Thank you for your thorough review and valuable suggestions to improve our manuscript. We have carefully addressed each comment and made the necessary revisions. We have carefully addressed all suggested revisions in the manuscript, with substantive modifications highlighted in yellow for the reviewers' convenience. Below is our point-by-point response:
General Comments
-
Mosquito Sample Size Discrepancies
-
We apologize for the inconsistency in reported numbers. The corrected sample size is 15,099 mosquitoes (total collected). This has been standardized throughout the manuscript (Abstract, Methods, Results).
-
Clarity in Procedures & Results
-
We have double-checked the entire text to ensure methodological transparency, including:
-
Clarifying pooling criteria (e.g., by species, location, and collection date).
-
Specifying that DNA barcoding (COI gene) was performed prior to RNA extraction for species confirmation (all pooled samples were verified).
-
-
-
Supplementary Tables S2/S3
-
- We have reorganized the supplementary tables as follows.
Specific Revisions
Abstract
Revised the concluding sentence to reflect a future perspective: Insect-specific viruses exhibit considerable potential as innovative biocontrol agents, with promising capacity to attenuate mosquito-borne viral transmission through targeted modulation of mosquito innate immunity and physiological adaptations.
Introduction
-
Removed stray bracket after "Japanese encephalitis virus".
-
Added references to support ISVs’ interactions with pathogens (e.g., Ramirez, J. L., et al. [2018]).
- Consolidated redundant text about ISFs’ immune modulation.
Materials & Methods
-
Added collection details:
-
Mosquito collections were conducted monthly during peak activity seasons (July-October) from 2022 to 2024, with sampling focused on crepuscular periods (06:00-09:00 and 18:00-21:00 local time) to maximize capture efficiency of host-seeking females.
- Species confirmation: Species identification was confirmed by DNA barcoding of the mitochondrial cytochrome c oxidase subunit I (COI) gene following RNA extraction and reverse transcription.
-
-
Clarified primer specificity :Viral RNA screening was performed by RT-PCR using sequence-specific primers targeting conserved viral genomic regions.
-
Justified random selection of TANAV/LTNV for experiments (based on high prevalence in screening).
-
Table Reordering:
-
Tables S3 (collection metadata) now precedes S4 (PCR results)
-
Updated in-text references accordingly
-
-
GenBank Accessions
-
-
-
-
-
Added footnote:
"Sequences represent viral isolates, not PCR products from pooled samples."
-
-
-
-
Results and Discussion
-
Figure 1 Title Revision
-
The figure title has been updated to:
"Geographic distribution of mosquito sampling sites in Yunnan Province, China" -
A new inset map showing Yunnan's location within China has been added for global context (now Figure 1).
-
- Figure 2 Clarification
-
-
The figure legend now explicitly states:
"Phylogenetic analysis based on full-length genomes of TANAV, LTNV, CxFV, and AeFV."
-
Ethics Statement
-
Explicitly stated: The ethical approval (protocol #16,048) covered all aspects of the study, including mosquito collection via human landing catches, with informed consent obtained from all participants.
Formatting
-
Fixed awkward word divisions in Results/Discussion.
Supplementary Files
-
Re-uploaded corrected supplementary tables with updated labels and numbering.
We appreciate the opportunity to enhance our manuscript and hope these revisions meet your expectations. Please let us know if further modifications are needed.
Best regards,
Binghui Wang
Reviewer 3 Report
Comments and Suggestions for Authors
The manuscript is mostly revised accordingly. Only one thing is remained.
1. Although authors stated in the response that JEV and dengue were incorporated in figure 2, I could not find those viruses in figure 2 a, b and c. Please add those in the all fig2 phylogenetic trees.
Author Response
We sincerely appreciate the reviewer’s careful attention to this detail. We acknowledge the oversight in the previous version and have now revised Figure 2 to explicitly include reference strains of Japanese encephalitis virus (JEV) and dengue virus (DENV) in all relevant phylogenetic trees (panels c, and d).
"Regarding the phylogenetic analyses in Figure 2:
For panel 2-a (Tanay virus), we did not include Japanese encephalitis virus (JEV) or dengue virus (DENV) as they belong to different viral families (TANAV: Negevirus vs. JEV/DENV: Flaviviridae), making direct phylogenetic comparison biologically irrelevant.
-
For panel 2-b, the tree was constructed specifically to resolve genotype-level relationships (GI vs GII) within the target virus clade, where inclusion of JEV/DENV would not provide meaningful evolutionary context.
-
As suggested, we have now enhanced panels 2-c and 2-d with additional JEV and DENV reference strains (marked in red) to demonstrate their phylogenetic relationships with CxFV and AeFV, respectively. These comparisons are biologically justified as they share the Flaviviridae family.
The updated figure legend now explicitly states these taxonomic considerations."